# The Role of Fatty Acid-Binding Protein 4 in the Characterization of Atrial Fibrillation and the Prediction of Outcomes after Catheter Ablation

**DOI:** 10.3390/ijms231911107

**Published:** 2022-09-21

**Authors:** José Nicolás López-Canoa, Marinela Couselo-Seijas, Teba González-Ferrero, Cristina Almengló, Ezequiel Álvarez, Adrián González-Maestro, Laila González-Melchor, José Luis Martínez-Sande, Javier García-Seara, Jesús Fernández-López, Bahij Kreidieh, Eva González-Babarro, José Ramón González-Juanatey, Sonia Eiras, Moisés Rodríguez-Mañero

**Affiliations:** 1Cardiovascular Department, Hospital Complex of Pontevedra, 36071 Pontevedra, Spain; 2Cardiology Translational Group, Health Research Institute of Santiago de Compostela, 15706 Santiago de Compostela, Spain; 3Department of Medicine, University of Santiago de Compostela, 15706 Santiago de Compostela, Spain; 4Cardiovascular Department, Hospital Complex of Santiago de Compostela, 15706 Santiago de Compostela, Spain; 5CIBERCV, Institute of Health Carlos III, 28220 Madrid, Spain; 6Beth Israel Deaconess Medical Center, Harvard Medical School, Boston, MA 02115, USA

**Keywords:** atrial cardiomyopathy, low-voltage areas, adiposity, fibrosis, catheter ablation

## Abstract

Aims: The utility of biomarkers in characterizing atrial cardiomyopathy is unclear. We aim to test the ability of biomarkers of fibrosis (galectin-3 (Gal-3)) and adiposity (fatty acid-binding protein 4 (FABP4) and leptin) to predict: (1) the presence of low-voltage areas (LVA) in the electroanatomic voltage mapping; and (2) the recurrence of atrial fibrillation (AF) after pulmonary vein isolation (PVI). Methods: Patients referred for PVI were enrolled. Areas of bipolar voltage < 0.5 mV were considered as LVA. An aggregate score incorporating AF pattern (paroxysmal, persistent and long-standing persistent) and peripheral levels of FABP4 (>20 ng/mL) was developed. Results: 299 patients were included. AF was paroxysmal in 100 (33%), persistent in 130 (43%) and long-standing persistent in 69 (23%). Multivariable analysis revealed age, left atrium area, and the proposed score as independent predictors of LVA. During a mean follow-up period of 972 ± 451 days, freedom from AF recurrence was 63%. The score incorporating AF pattern and FABP4 levels accurately predicted freedom from AF recurrence, stratifying risk into ranges from 28% (score of 1) to 68% (score of 3). Cox regression models identified the score including AF pattern + FABP4 as the best model for AF recurrence (hazard ratio 2.32; 95% CI, 1.19 to 4.5; *p* = 0.014). Conclusions: Traditional clinical classification of atrial cardiomyopathy may be improved by markers of adiposity (FABP4). The combination allows better prediction of the presence of LVA and AF recurrence post-PVI. Gal-3 provided no added predictive value.

## 1. Introduction

Various pathological processes affecting the atria contribute to onset, development and perpetuation of atrial fibrillation (AF). ‘Atrial cardiomyopathy’ refers to the pathophysiological changes that affect the atria with possible clinical implications. An expert consensus has proposed a histological and pathophysiological classification of atrial cardiomyopathy based on predominant pathologic features [1]. However, at the present time, it is difficult to ascertain the best way to characterize atrial cardiomyopathy. Epicardial adipose tissue (EAT) and fibrosis in the left atrium (LA) can be detected and evaluated as low-voltage areas (LVA) at the time of invasive electroanatomic voltage mapping (EAM) [2]. Although the measured voltage is influenced by several factors, LVA assessment can be performed below specific selected thresholds [3]. LVA assessment is useful, first, as a modulator of the ablation strategy to be implemented, and second, as a marker of success after ablation procedures. Jadidi et al. have proposed that pulmonary vein isolation (PVI) alone is sufficient if the identified LVA do not exceed 10% of the left atrial (LA) surface area [4]. A major limitation of this strategy is that this information is not available until the time of the procedure, and other noninvasive methods for determining the extent of atrial cardiomyopathy remain limited. Alternatively, several biomarkers have been proposed to reflect the level of atrial remodeling. Galectin-3 (Gal-3), a member of the beta-galactosidine-binding lectin family, is a biomarker related to tissue fibrosis and inflammation that has been associated with atrial fibrosis and disease progression in AF. Fatty acid-binding protein 4 (FABP4) is a master regulator of the transcription factor peroxisome proliferator-activated receptor gamma involved in adipocyte differentiation [5] and is thus a marker of adipose tissue inflammatory activity [6]. Leptin, an anorexic hormone released by adipocytes, regulates the pathogenesis of atrial fibrosis evoked by angiotensin II. Both adipokines are co-regulated during the inflammatory process of adipose tissue [7], and, interestingly, the plasma levels of both proteins differ between men and women depending on AF burden [8]. At present, conflicting evidence exists regarding the utility of biomarkers in detecting atrial cardiomyopathy as well as their role in predicting arrhythmia recurrence after AF ablation.

We aim (a) to test the ability of circulating biomarkers of fibrosis (Gal-3) and adiposity (FABP4 and leptin) to predict the presence of LVA in the EAM and (b) to assess their role in the prediction of AF recurrence after PVI.

## 2. Results

### 2.1. Population Characteristics

A total of 299 patients, with a mean age of 58 ± 10 years, 98 (33%) of whom were women, were included. AF was paroxysmal in 100 (33%), persistent in 130 (43%) and long-standing persistent in 69 (23%) patients. The mean body mass index (BMI) was 29.7 ± 4.8 kg/m^2^. The average LA area was 20.4 ± 6 cm^2^. The median and interquartile range of Gal-3 levels were similar between atrial (8.6; 5.8–12.9 ng/mL) and peripheral (8.5; 5.7–13.7 ng/mL) samples. The median and interquartile range of FABP4 levels were higher in peripheral (19.6; 12.4–29.2 ng/mL) than in atrial samples (15.9; 9.6–24.7 ng/mL). Most patients were taking oral anticoagulation (direct oral anticoagulants 62% vs. vitamin K antagonists 36%). About 69% of patients were receiving beta-blockers at the time of the ablation. There were some differences between males and females. Females were significantly older than males, with similar BMI but differences in FABP4 and leptin levels (see Appendix A). Mean follow-up was 972 ± 451 days, and 258 patients completed more than 12 months of follow-up. The main baseline characteristics are shown in Table 1.

### 2.2. Extent of Low-Voltage Area on EAM

In our population, LVA comprised a median of 1.44% (interquartile range 0.06–9.98). Out of the 299 patients, 66 (22.1%) had LVA ≥ 10%. In the univariable analysis, age, arterial hypertension, diabetes mellitus, LA area, low-density lipoprotein cholesterol, triglycerides (TG), and the proposed score were associated with the presence of LVA. However, multivariable analysis revealed that only age, LA area, and the proposed score were independent predictors of LVA (Table 2). According to this score (based on AF type and FABP4 levels), our population was divided into three categories (score of 1, score of 2, or score of 3). The basic clinical data of the patients in these three groups are shown in Table 3. The group with the highest score had a higher BMI, more extensive LVA, larger LA area, lower LVEF, and a higher percentage of patients with arterial hypertension.

### 2.3. Recurrence after AF Catheter Ablation

Freedom from AF recurrence in our population was 63% during the follow-up period. As shown in the Kaplan–Meier curves, AF recurrence was detected in 28% of patients assigned a score of 1, in 41% of patients assigned a score of 2, and in 68% of patients assigned a score of 3 (log rank *p* < 0.0001), (Figure 1A). However, if we only take into account the type of AF, recurrence was detected in 26% of patients with paroxysmal AF, in 36% of patients with persistent AF, and in 54% of patients with long-standing persistent AF (Figure 1B). The addition of FABP4 levels to the clinical model (only AF type) reduced the Akaike Information Criterion (AIC) value (from 1146.18 to 1141.28), indicating that stratifying patients by AF type and FABP4 levels (the proposed score) reclassified them in a more efficient manner than the traditional AF clinical classification. Specifically, cox regression models identified the score of 3 (referred to long-standing persistent AF with FABP4 > 20 ng/mL) with the highest risk for AF recurrence (hazard ratio 2.32; 95% CI, 1.19 to 4.5; *p* = 0.014), as shown in Table 4.

## 3. Discussion

The present study aims to perform a clinical and biomarker analysis of patients with AF referred for PVI and the associated impact on clinical outcomes. The main finding of the study is that the traditional clinical classification of AF (paroxysmal, persistent and long-standing persistent) can predict the long-term results of AF catheter ablation. However, this predictive scheme can be augmented by markers of adiposity (FABP4), which can be of particular interest in patients with a persistent pattern. These findings might be useful in optimizing patient selection for ablation. Moreover, they may contribute to a better understanding of the mechanism leading to AF and potentially to the development of alternative treatment strategies.

### 3.1. Atrial Fibrillation Patterns

Different AF classifications have been proposed and, traditionally, five patterns of AF are distinguished, based on presentation, duration, and spontaneous termination of AF episodes. Often, a more global classification is applied, discriminating only paroxysmal from non-paroxysmal AF [9]. The transition from paroxysmal to non-paroxysmal AF is often characterized by advanced atrial structural remodeling [10,11]. The variability seen in AF ablation success rates may in part be due to these atrial morphofunctional changes, particularly in patients with persistent AF. Conversely, atrial cardiomyopathy can sometimes be present in patients with paroxysmal AF. As such, the standard clinical classification may not reliably reflect the pathophysiologic process underlying AF. Current guidelines thus emphasize that more data is needed to define the prognostic and treatment implications of different atrial cardiomyopathy subtypes. Within this context, the added value of relevant biomarkers is not yet well defined.

### 3.2. Atrial Cardiomyopathy and Biomarkers

The exact mechanism of the origin and progression of LA fibrosis in human AF is not precisely known. Fibrosis is a well-known factor that plays a role in AF pathophysiology [12]. It has been shown to induce an arrhythmogenic substrate by invoking new micro re-entry circuits, electrical heterogeneity, regions of local conduction block, and alterations in atrial refractory periods. Although it seems to be an independent predictor of catheter ablation failure, considerable controversy still persists.

In the clinical setting, a diagnosis of atrial myopathy/LA fibrosis can be discerned by advanced ECG signal processing, echocardiographic imaging of LA geometry and magnetic resonance imaging of myocardial fibrosis [13]. At the time of AF ablation, the presence of atrial cardiomyopathy can be inferred by means of the EAM. Although various ranges of voltage values have been reported, it is generally agreed upon that voltage values below 0.5 mV in sinus rhythm are suggestive of the presence of an underlying fibrotic scar [14]. Thus, the mapping of low-voltage values as surrogates for fibrosis represents an attractive and seemingly straightforward means of interrogating the atrial substrate (Figure 2). The APPLE score has been shown to predict LVA in the atria, which represents advanced remodeling processes associated with higher rates of arrhythmia recurrence [15].

One of the factors influencing the presence of LVA in non-valvular AF is the amount of EAT [16]. EAT is a heterogenous microenvironment composed of adipocytes, fibroblasts, immune cells, nerve cells and blood vessels that is on paracrine cross-talk with the myocardium. The influence of EAT on arrhythmogenesis is multifactorial [17]. Directly, fatty infiltration is an anatomic obstacle to cardiac excitation that delays conduction. Secondly, EAT expansion is linked to lipid overload, leading to calcium dysregulation, mitochondrial dysfunction, oxidative stress, and inflammation. Importantly, chronic inflammatory and metabolic disorders are accompanied by an expansion of EAT, and the increase in EAT mass is proportional to the clinical severity of the disease and the intensity of systemic inflammation [18]. For these reasons, any latent inflammation-related disease (infectious, autoimmune, oncological, etc.) was excluded in this study. Thirdly, adipokines secreted by EAT induce ion channel remodeling, gap junction remodeling, and atrial fibrosis (due to proinflammatory autocrine and paracrine secretion). However, despite this known mechanistic role, there are no well-established markers of EAT activity at the clinical level. FABP4, also called adipocyte protein 2, is an adipokine involved mainly in the trafficking of fatty acids and other lipophilic molecules, and in the differentiation of adipocytes. It is also frequently expressed in macrophages, where it appears to regulate redox signaling and to promote inflammatory response. Higher levels of FABP4 have been shown to depend on AF burden, and proteomic studies have identified FABP4 as a new risk marker for AF [19]. Its association with the development of fibrosis was first described in the kidney, and its pharmacological inhibition counteracted renal dysfunction. A possible explanation could be the attenuation of the macrophage–myofibroblast transition, induced by FABP4 [20]. Subsequently, FABP4 has been linked to structural heart disease and contractile dysfunction by modulating calcium dynamics in cardiomyocytes [21]. Our results demonstrate that its levels were independent predictors of LVA and success rates after PVI. These findings suggest that FABP4 may play an important role in the proarrhythmogenic effects of EAT. Relevantly, it might be a useful marker of atrial pathological remodeling. However, further mechanistic studies are needed to understand this effect and the link between FABP4, atrial fibroblasts, and atrial cardiomyopathy.

On the other hand, compared to men, women had higher levels of FABP4. Although the exact mechanisms remain elusive, clinical investigations have indicated sexual dimorphism in AF. Consistent with this rationale, in our previous study we have found an association between higher levels of FABP-4 in women with AF compared to men [8]. Recently, Ruben R. De With et al. have shown that FABP4 levels were higher in women than in men with paroxysmal AF [22]. Based on the fact that FABP4 can provide information on the amount of body fat, this could be an argument to explain the higher levels in women. However, we did not obtain body fat percentage by dual-energy X-ray absorptiometry. Moreover, sex-related differences between sex hormones and adipose tissue secretion have also been described [23]. Thus, with the same amount of visceral fat, there are differences in the secretome of adipose tissue between women and men, which could explain, at least in part, the significant sex differences detected in patients with AF. In addition, another reason that may explain this difference is the fact that in our population the vast majority of women have experienced menopause at the time of ablation. Related to hormonal change, FABP4 levels increase considerably after menopause [24]. Therefore, the conclusions of the present study may not be applicable to non-menopausal women. Further investigation into factors underlying sex-based differences may provide mechanistic insight into AF development.

### 3.3. Clinical Relevance

All in all, our results showed that both FABP4 levels and AF type might identify the presence of LVA, a major feature of the progression of AF. The proposed score, which combines AF type along with peripheral levels of FABP4, permits reclassifying patients with AF into low- or high-risk groups for AF recurrence post-PVI. Since paroxysmal AF pattern falls exclusively within Score 1, our score mainly assesses non-paroxysmal AF patterns (persistent and long-standing persistent AF). In fact, we have identified a subgroup of patients (long-standing persistent AF with FABP4 > 20 ng/mL) in which the success rate was dismal (below 35%). As such, the study’s findings might be of interest not only for optimized patient selection for AF ablation, but also for the identification of patients who may benefit from closer follow-up. Potentially, these results could lead to further studies investigating the role of personalized therapies such as FABP4 modifiers in AF patients.

### 3.4. Study Limitations

Several limitations should be considered. The study has the inherent limitations of a non-randomized retrospective study with a small number of patients in a single center. The lack of significant association with a clinical variable or biomarker could in fact be due to a lack of statistical power. Importantly, our main purpose was to characterize AF patients and to investigate the relationship between different biomarkers with the presence of LVA in the EAM at the time of AF ablation. Although we also investigated the trend of these biomarkers with ablation outcomes, the two aren’t necessarily intrinsically linked. In patients who have a long-standing systemic inflammatory or metabolic disorder responsible for the atrial myopathy, the pathophysiologic process in the atria might not necessarily resolve through treatment of AF. On the other hand, AF recurrence after ablation could be due to specific procedural factors (such as the presence of gaps) as opposed to patient-specific preprocedural aspects. In order to counteract this bias, we only included patients referred to point-by-point radiofrequency ablation with the same technology, incorporating contact-force-sensing technology in all of them. MRI has been shown to be a non-invasive and accurate tool for assessing atrial myopathy; unfortunately, it was not performed for the sake of the present study, so information about its correlation with the analyzed biomarkers and LVA cannot be deduced. Nearly 16% of patients included underwent previous PVI. It is known that RF lesions could interfere with the level of myopathy found on EAM. For this reason, patients with additional lesions beyond the PV were not included in the analysis. Moreover, LVA quantification was performed outside the PV to minimize any confounding effect resultant from prior ablation effects. Finally, AF burden was determined by temporal pattern and intermittent ECG monitoring, neither corresponding well to the long-term ECG monitoring.

## 4. Methods and Materials

### 4.1. Subjects

From September 2016 till September 2020, consecutive patients with paroxysmal, persistent or long-standing persistent AF, classified according to clinical practice guidelines [9], and referred for PVI in a single center, were included in the study. The exclusion criteria were age under 18 years, pregnancy and any latent infectious condition. Patients with previous PVI were only included if no additional lesion beyond the PV was previously performed. The study protocol was approved by the Ethical Committee of Clinical Research of Santiago-Lugo (identification code: 2019_439). Written informed consent was obtained from all participants.

### 4.2. Blood Sample Collection

Prior to the PVI (in the fasted state), just after the transeptal puncture, and previous to heparin administration, blood sample were obtained from the LA through the transeptal sheath. At the same time, a peripheral blood sample was drawn from the antecubital vein using an 18-G butterfly cannula with a two-syringe technique, discarding the first 5 mL and using the second 5 mL for testing [25]. Blood samples were collected in EDTA-tubes.

### 4.3. Plasma Measurements

Plasma samples were centrifuged and stored at −80 °C until use. Leptin and FABP4 levels were determined using a magnetic Luminex multiplex test kit (R&D Systems, Minneapolis, MN, USA). The sensitivity for Leptin and FABP4 was 10.2 and 95.7 pg/mL, respectively. An enzyme-linked immunosorbent assay (ELISA) kit (BMS279-4; eBioscience, Vienna, Austria) was used to analyze Gal-3 levels. Measurements were performed in duplicate, and the results were averaged. The intra-assay and inter-assay coefficients of variation were 7.5% and 5.4%, respectively.

### 4.4. Ablation Procedure and Patient Follow-Up

We included patients who underwent point-by-point radiofrequency PVI using Ablation Index (SmartTouch, Biosense Webster, Inc., Diamond Bar, CA, USA). The majority of patients were discharged 24–36 h after the ablation procedure. Systemic anticoagulation was continued for at least 2 months post ablation in all patients, and oral anticoagulation was continued long term in those patients with a CHA_2_DS_2_-VASc score of ≥1 in men or ≥2 in women. In our center, it is standard of care to continue antiarrhythmic drug therapy (ADT) during the blanking period (defined as 3 months after ablation). If patients remain free of recurrence after these 3 months, they are encouraged to stop ADT and only restart it in case of recurrence. In case of a second recurrence beyond the blanking period, and after resuming ADT or electrical cardioversion, patients are considered for a redo ablation procedure. Patients were systematically reviewed at 3, 6, 12 and 24 months after the index procedure. Each medical visit included a detailed medical history, physical exam and 12-lead electrocardiogram (ECG). Monitoring with intermittent Holter-ECG was at the physician’s discretion. Recurrence was defined as ECG documented AF, atrial flutter or atrial tachycardia >30 s in duration and occurring after the blanking period.

### 4.5. Left Atrial Low-Voltage Areas

A bipolar voltage map was created simultaneously with LA surface reconstruction, guided by an EAM system (CARTO3, Biosense Webster, Inc., Diamond Bar, CA) using a multipolar mapping catheter (PentaRay, Biosense Webster, Inc., Diamond Bar, CA). Patients in AF rhythm at the start of the procedure systematically underwent electrical cardioversion in the electrophysiology laboratory. Adequate quality of the acquired voltage points was established according to the CONFIDENSE module after respiratory compensation. This is continuous mapping software with automated data acquisition when set criteria are met, among them: (1) tissue proximity indication; (2) wavefront annotation; (3) map consistency; (4) position stability filter; (5) cycle length stability (keeps data collected within a range of predefined cycle lengths, within 10% of the average). A minimum number of points was requested (>1000) and the density fill threshold remained constant at ≤5 mm.

Contiguous areas of bipolar voltage < 0.5 mV were considered as LVA in sinus rhythm [14]. Total LA surface area was defined as the LA body area without the PV antrum regions, LA appendage orifice, and mitral valve. Medians of the total LA surface area and area of each predefined region were measured offline on the three-dimensional reconstructed LA model. Median values of LVA were set in relation to the surface area of each region and the entire LA.

### 4.6. AF Type and FABP4 Score

An aggregate score was assigned. It was established based on the type of AF and the peripheral levels of FABP4: A score of 1 is referred to paroxysmal AF (independently FABP4 levels) or to persistent AF with FABP4 ≤ 20 ng/mL. A score of 2 is referred to persistent AF with FABP4 > 20 ng/mL or to long-standing persistent AF with FABP4 ≤ 20 ng/mL. A score of 3 is referred to long-standing persistent AF with FABP4 > 20 ng/mL. The cut-off point of 20 ng/mL for FABP4 levels was chosen because it is the median level (19.6 ng/mL) in the study population.

### 4.7. Statistical Analyses

Categorical variables were represented as percentage and continuous variables were represented as mean ± standard deviation (SD) or interquartile range according to their normal or scatter distribution. Time 0 corresponds to the moment of PVI. Comparison of survival curves amongst scores was performed using a log-rank test. We use AIC model selection to distinguish the best-fit model. A cox proportional hazard model was used to evaluate the effect of these scores on freedom from AF recurrence. All analyses were programmed in R 3.5 (R Core Team, Vienna, Austria), and *p* < 0.05 was considered statistically significant.

## 5. Conclusions

It is already well documented that the traditional clinical classification of AF (paroxysmal, persistent and long-standing persistent) predicts the long-term results of AF ablation. However, the main finding of our study is that this classification may be improved by incorporating adiposity markers (FABP4), which correlate with the presence of LVA in EAM. These findings are meaningful not only for optimizing the selection of patients referred for ablation, and particularly within the long-standing persistent AF subtype with high-levels-of-FABP4 subgroup with a high probability of AF recurrence, but also for a better understanding of the mechanism leading to AF cardiomyopathy.

## Figures and Tables

**Figure 1 ijms-23-11107-f001:**
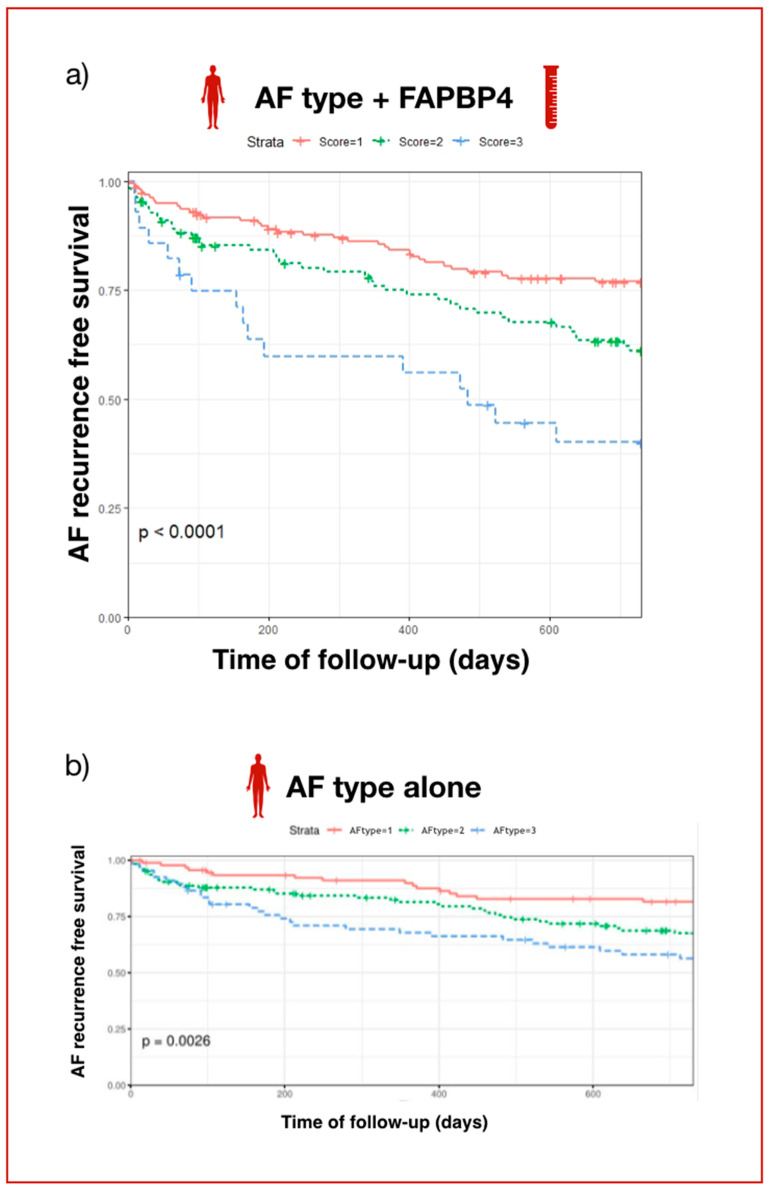
Kaplan–Meier curves estimating freedom from atrial fibrillation after pulmonary vein isolation: (**a**) considering AF type and FABP4 levels (proposed score). (**b**) considering only AF type (1: paroxysmal; 2: persistent; 3: long-standing persistent).

**Figure 2 ijms-23-11107-f002:**
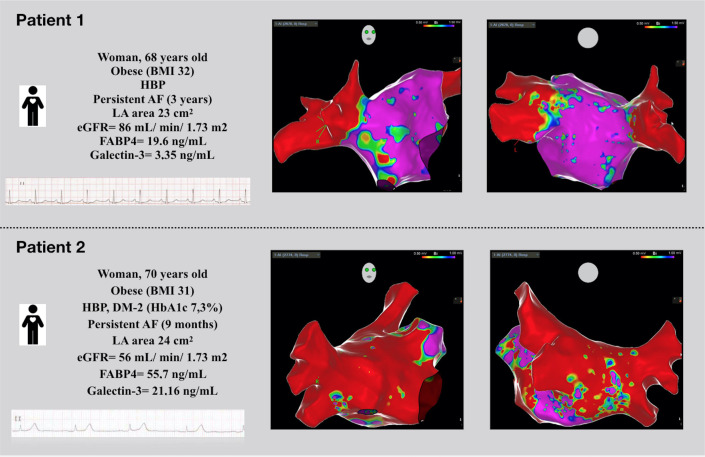
Three-dimensional electroanatomical mapping during catheter ablation. Patient 1: 68 years old, female, with persistent AF, non-elevated FABP4 levels and isolated atrial fibrosis. No recurrence of AF. Patient 2: 70 years old, female, with persistent AF, elevated FABP4 levels and extensive atrial fibrosis. Early recurrence of AF.

**Table 1 ijms-23-11107-t001:** Characteristics of the patients at baseline.

	%	N	Mean	SD	Percentiles		
					25	50	75
Gender (female/male)	33/67	98/201					
Age		299	58.2	10	52	59	66
BMI (Kg/m^2^)		299	29.7	4.8	26.7	29.4	32
AF type (1/2/3)	33/43/23	100/130/69					
LA LVA (%)		299	9.6	17.8	0.06	1.44	9.98
Years (AF)		299	3.8	4.6	1	2	5
Redo procedure (no/yes)	11	266/33					
Heart Rate (bpm)		299	70	21.5	56	65	80
PR (ms)		299	161.4	31.5	140	160	180
QRS (ms)		299	92.7	15.4	80	90	100
LA area (cm^2^)		299	20.4	6	15	20	24
LVEF		299	60.6	9	56	61	66
Laboratory measurements							
Hemoglobin (g/dL)		299	14.3	1.4	13.4	14.4	15.3
Platelets (10^9^/L)		299	201.9	50.5	170	200	235
Blood urea nitrogen (mg/dL)		299	47	15.7	38	45	53
Creatinine (mg/dL)		299	1.3	3.4	0.8	1	1.1
eGFR (mL/min/1.73 m^2^)		299	95.8	32.5	72.5	90.9	114.7
Plasma Sodium (mEq/L)		299	140.6	8.9	140	141	143
Plasma Potassium (mEq/L)		299	4.4	2.3	4	4.2	4.5
Total cholesterol (mg/dL)		299	189	39.2	163	188	215
LDLc (mg/dL)		299	115.1	39.2	163	188	215
HDLc (mg/dL)		299	51.4	15.8	41.7	50	59
Triglycerides (mg/dL)		299	118	57.6	84	105	138.3
Glucose (mg/dL)		299	106.7	24.3	93	103	113
HbA1c (g/dL)		199	5.7	0.6	5.3	5.6	5.9
TSH (mU/L)		299	2.9	2.3	1.6	2.3	3.6
LA Gal-3 (ng/mL)		272	10.1	6.3	5.8	8.6	12.9
Peripheral Gal-3 (ng/mL)		299	11.5	6.9	5.7	8.5	13.7
LA FABP4 (ng/mL)		274	20.5	16.7	9.6	15.9	24.7
Peripheral FABP4 (ng/mL)		299	24.1	18.2	12.4	19.6	29.2
LA Leptin (ng/mL)		271	21.2	27.8	6.5	12.4	24.6
Peripheral Leptin (ng/mL)		299	23.9	32.2	8.5	17.21	31.4
Disease-Risk factors							
Taquicardiomyopathy (no/yes)	16	251/48					
AHT (no/yes)	46	161/138					
T2DM (no/yes)	13	260/39					
Smoker (no/yes)	30	209/90					
COPD (no/yes)	7	281/18					
OSA (no/yes)	5	284/15					
Treatments							
Statins (no/yes)	43	170/129					
ACEi (no/yes)	20	239/60					
ARB (no/yes)	23	230/69					
NDHP CCB (no/yes)	5	284/15					
Vitamin K antagonist (no/yes)	36	191/108					
DOAC (no/yes)	62	114/185					
Class I ADT (no/yes)	32	203/96					
Class II ADT (no/yes)	69	93/206					
Class III ADT (no/yes)	29	212/87					
Class IV ADT (no/yes)	7	278/21					
Follow-up (days)		299	972	451			

BMI: Body Mass Index; AF type: Atrial Fibrillation type (1: paroxysmal; 2: persistent; 3: long-standing persistent); LA: Left atrium; LVA: Low-voltage areas; LVEF: Left Ventricular Ejection Fraction; eGFR: Estimated Glomerular Filtration Rate; Gal-3: Galecin-3; FABP4: Fatty Acid-Binding Protein 4; AHT: Arterial Hypertension; T2DM: Type 2 Diabetes Mellitus; COPD: Chronic Obstructive Pulmonary Disease; OSA: Obstructive Sleep Apnea; ACEi: Angiotensin-Converting Enzyme inhibitors; ARB: Angiotensin Receptor Blockers; NDHP CCB: Nondihydropyridine Calcium Channel Blockers; DOAC: Direct Oral Anticoagulants; ADT: Antiarrhythmic Drug Therapy.

**Table 2 ijms-23-11107-t002:** Multivariable linear regression analysis: low-voltage area as dependent variable.

	Coefficient	CI95%	Sig.
Age	0.262	0.03–0.49	0.028
AHT	3.084	−1.81–7.98	0.218
T2DM	4.233	−2.99–11.46	0.252
LA area	0.558	0.18–0.94	0.005
LDLc	−0.031	−0.10–0.04	0.396
TG	−0.035	−0.07–0.003	0.072
Score of 1			
Score of 2	2.557	−2.13–7.24	0.286
Score of 3	10.97	3.37–18.56	0.005

AHT: Arterial Hypertension; T2DM: Type 2 Diabetes Mellitus; LA: Left atrium; LDLc: low-density lipoprotein cholesterol; TG: Triglycerides.

**Table 3 ijms-23-11107-t003:** Clinical characteristics of patients regarding score (AF type plus FABP4 levels).

	Score 1 (*n* = 162)	Score 2 (*n* = 112)	Score 3 (*n* = 28)	Sig. (*p* Value)
Gender (female *n*/%)	52/32	32/28	14/50	0.1304
Age	58 (52–66)	59(52–66)	63 (55–70)	0.1168
BMI (Kg/m^2^)	29 (26–32)	29(27–30)	31 (28–34)	0.0329
AF type (1/2/3)	100/62/0	0/70/42	0/0/28	
LVA (%)	1 (0.06–6.78)	3.4(0.1–10)	7 (0.25–35)	0.0274
PR (ms)	163 ± 33	159 ± 28	157 ± 17	0.9410
QRS (ms)	93 ± 14	92 ± 17	93 ± 17	0.4270
LA area (cm^2^)	19 ± 6	21 ± 6	23 ± 6	0.0010
LVEF	63 ± 8	59 ± 9	59 ± 12	0.0010
Laboratory measurements				
Hemoglobin (g/dL)	14 ± 2	14 ± 1	14 ± 1	0.1378
Platelets (10^9^/L)	206 ± 52	198 ± 48	199 ± 40	0.4383
Blood urea nitrogen (mg/dL)	46 ± 12	47 ± 16	50 ± 15	0.4216
Creatinine (mg/dL)	1.0 ± 0.2	1.4 ± 0.4	1.0 ± 0.3	0.7110
eGFR (mL/min/1.73 m^2^)	96 ± 30	95 ± 34	94 ± 55	0.7556
Plasma Sodium (mEq/L)	141 ± 10	141 ± 5	142 ± 2	0.2247
Plasma Potassium (mEq/L)	4.2 ± 0.4	6.3 ± 0.1	4.2 ± 0.3	0.0379
Total cholesterol (mg/dL)	190 ± 39	188 ± 40	185 ± 35	0.7170
LDLc (mg/dL)	116 ± 32	116 ± 32	108 ± 28	0.3504
HDLc (mg/dL)	52 ± 17	50 ± 14	54 ± 13	0.4501
Triglycerides (mg/dL)	116 ± 51	118 ± 52	131 ± 99	0.9708
HbA1c (g/dL)	5.7 ± 0.5	6.4 ± 0.6	5.8 ± 0.7	0.5504
LA Gal-3 (ng/mL)	8.5 (5.7–13)	8.8 (5.8–13)	7.8 (6.0–10)	0.4687
Peripheral Gal-3 (ng/mL)	8.5 (5.7–14)	8.8 (5.8–14)	7.6 (5.5–11)	0.6107
LA FABP4 (ng/mL)	13 (8.7–18.5)	20 (11–29)	24 (21–28)	<0.001
Peripheral FABP4 (ng/mL)	16 (11–22)	24 (13–36)	28 (24–38)	<0.001
Disease and risk factors				
AHT (*n*/%)	52/32	44/39	19/68	0.0178
T2DM (*n*/%)	19/11	13/12	6/21	0.3914
Smoker (*n*/%)	51/31	37/33	4/14	0.4800
COPD (*n*/%)	9/0.05	10/0.09	0/0	0.1926
OSA (*n*/%)	7/0.04	9/0.08	1/0.04	0.3933
Treatments				
Statins (*n*/%)	68/42	50/45	13/46	0.7821
ACEi (*n*/%)	27/17	24/21	8/28	0.2899
Class I ADT (*n*/%)	59/36	32/28	4/14	0.0374
Class II ADT (*n*/%)	100/62	87/78	23/82	0.0051
Class III ADT (*n*/%)	40/25	34/30	13/46	0.0798
Class IV ADT (*n*/%)	11/0.07	9/0.08	2/0.07	0.9455

BMI: Body Mass Index; AF type: Atrial Fibrillation type (1: paroxysmal; 2: persistent; 3: long-standing persistent); LA: Left atrium; LVA: Low-voltage areas; LVEF: Left Ventricular Ejection Fraction; eGFR: Estimated Glomerular Filtration Rate; Gal-3: Galecin-3; FABP4: Fatty Acid-Binding Protein 4; AHT: Arterial Hypertension; T2DM: Type 2 Diabetes Mellitus; COPD: Chronic Obstructive Pulmonary Disease; OSA: Obstructive Sleep Apnea; ACEi: Angiotensin-Converting Enzyme inhibitors; ADT: Antiarrhythmic Drug Therapy.

**Table 4 ijms-23-11107-t004:** Cox regression model with AF recurrence as dependent variable.

	HR (95% CI)	Sig.
Gender (male)	0,924 (0.591–1.445)	0.729
LA area	1.028 (0.991–1.066)	0.139
TG	0,997 (0.994–1,001)	0.179
LVA	1.014 (1.005–1.024)	0.003
Score of 1		
Score of 2	1.828 (1.170–2.860)	0.008
Score of 3	2.320 (1.190–4.524)	0.014

LA: Left atrium; TG: Triglycerides; LVA: Low-voltage areas.

## Data Availability

The data presented in this study are available on request from the corresponding author. The data are not publicly available due to ethical restrictions.

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
