# Peer review of "The Role of Fatty Acid-Binding Protein 4 in the Characterization of Atrial Fibrillation and the Prediction of Outcomes after Catheter Ablation"

_ijms, 2022, doi:10.3390/ijms231911107_

Round 1

Reviewer 1 Report

López-Canoa and colleagues present a manuscript on the association of biomarkers of fibrosis (galectin-3), adipose tissue inflammatory acitivty (fatty acid-binding protein 4), extent of left atrial low voltage areas and the outcome of catheter ablation of atrial fibrillation (AF) in a mixed cohort of 299 patients undergoing AF ablation.

The background and rationale of the study are well described. The topic is interesting and the manuscript is well written.

I have the following comments:

The authors mention, that the aim of their study is to test the ability of the above mentioned biomarkers to predict the extent of low voltage areas and the success rate of AF ablation.

In this context, I am missing a results section with positive and negative predictive values together with the 95% confidence intervals for the newly developed score to predict low voltage areas and success of catheter ablation.
Additionally, I would be interested in differences in prediction among the different AF patterns? In other words, is the new algorithm able to identify patients with paroxysmal AF with such low success rates of AF ablation, that an AF ablation procedure should be abandoned or does the new algorithm in particular help to better characterize patients with persistent and long standing persistent AF?

In the text of the results section, the authors report the median of left atrial low voltage, while in Table 1 they give the mean and the 50th percentile. This is confusing. I would also recommend to move this information to section 2.2. This paragraph should be more elaborate and not only give the results of univariate and multivariate association studies. For example, how many patients had low voltage < 10%? How was the distribution among the different AF patterns (paroxysmal, persistent, long standing persistent)?

Minor comments:

Page 1: Introduction instead of introduction

Table 1: tachycardiomyopathy instead of taquicardiomyopathy

Table 2: linear regression instead of lineal regression

I would recommend to combine Figure 1 a) and b) to one single graph with six lines so the outcomes of only AF pattern (now graph b) and the addition of FABP4 can be better appreciated.

Figure 2 nicely represents two case examples, success of PVI should also be reported.

Author Response

We would like to thank the reviewer for his effort in revising our manuscript, which have contributed considerably to strengthening our study. Kindly find all original reviewer comments followed by our response below. The changes implemented in the manuscript are highlighted in yellow.

Reviewer 2 Report

The authors presented that the Traditional clinical classification of atrial cardiomyopathy may be improved by markers of adiposity (FABP4). But on the other hand, the manuscript is not ready to submit. The Authors required an extensive submission format, for reasons below

1. We have already known that circulating FABP4 levels correlated positively with CVD, especially atherosclerosis and cardiovascular outcomes. High levels of FABP4 were observed in patients with CVD which might be implicated that FABP4 is not only potential biomarker for AF. recent research have reported that LVA do not absolutely reflect fibrosis (front Cardiovasc med, 2022). therefore, the authors need to have a concrete hypothesis and suggest a specific mechanism. for example on figure 2, Patient 2 has T2-DM which could be more associated with FABP4 than AF independent according to your figure.

2. the authors have to show gender differences and significance on tables. in addition, need to explain and specify AF1 to 3 for examples on the table, TIPOFA in the figures, where comes from significance in figure1.

3. No explanations and legend for Figures 1  and 2 

Author Response

(The authors gave the same response as above.)

Round 2

Reviewer 2 Report

Since the authors know that FABP4 levels were different in gender, How would you explain that they are predominantly higher in females than males, It would be better described in the discussion.

Author Response

(The authors gave the same response as above.)
